# Clinical Performance of Rapid and Point-of-Care Antigen Tests for SARS-CoV-2 Variants of Concern: A Living Systematic Review and Meta-Analysis

**DOI:** 10.3390/v14071479

**Published:** 2022-07-06

**Authors:** Jimin Kim, Heungsup Sung, Hyukmin Lee, Jae-Seok Kim, Sue Shin, Seri Jeong, Miyoung Choi, Hyeon-Jeong Lee

**Affiliations:** 1Division of Healthcare Technology Assessment Research, National Evidence-Based Healthcare Collaborating Agency, 400, Neungdong-ro, Gwangjin-gu, Seoul 04933, Korea; jimin@neca.re.kr (J.K.); mychoi@neca.re.kr (M.C.); leehj@neca.re.kr (H.-J.L.); 2Departments of Laboratory Medicine, Asan Medical Center, University of Ulsan College of Medicine, 88, Olympic-ro 43-gil, Songpa-gu, Seoul 05505, Korea; sung@amc.seoul.kr; 3Departments of Laboratory Medicine, Yonsei University College of Medicine, 50-1, Yonsei-ro, Seodaemun-gu, Seoul 03722, Korea; hmlee71@yuhs.ac; 4Department of Laboratory Medicine, Kangdong Sacred Heart Hospital, Hallym University College of Medicine, 150, Seongan-ro, Gangdong-gu, Seoul 05355, Korea; jaeseokcp@gmail.com; 5Departments of Laboratory Medicine, Seoul Metropolitan Government-Seoul National University Boramae Medical Center, 20, Boramae-ro 5-gil, Dongjak-gu, Seoul 07061, Korea; jeannie@snu.ac.kr; 6Departments of Laboratory Medicine, Kangnam Sacred Heart Hospital, Hallym University College of Medicine, 1, Singil-ro, Yeongdeungpo-gu, Seoul 07441, Korea

**Keywords:** SARS-CoV-2, COVID-19, variant, rapid antigen test, performance

## Abstract

Rapid antigen tests (RATs) for detecting severe acute respiratory syndrome coronavirus 2 (SARS-CoV-2) are widely used in the Coronavirus disease 2019 (COVID-19) pandemic caused by diverse variants. Information on the real-world performance of RATs for variants is urgently needed for decision makers. Systematic searches of the available literature and updates were conducted in PubMed, Ovid-MEDLINE, Ovid-EMBASE, CENTRAL, and KMBASE for articles evaluating the accuracy of instrument-free RATs for variants up until 14 March 2022. A bivariate random effects model was utilized to calculate pooled diagnostic values in comparison with real-time reverse transcription-polymerase chain reaction as the reference test. A total of 7562 samples from six studies were available for the meta-analysis. The overall pooled sensitivity and specificity of RATs for variants were 69.7% (95% confidence interval [CI] = 62.5% to 76.1%) and 100.0% (95% CI = 98.8% to 100.0%), respectively. When an additional 2179 samples from seven studies reporting sensitivities only were assessed, the pooled sensitivity dropped to 50.0% (95% CI = 44.0% to 55.0%). These findings suggest reassessment and monitoring of the diagnostic utility of RATs for variants, especially for the sensitivity aspect, to facilitate appropriate diagnosis and management of COVID-19 patients.

## 1. Introduction

Coronavirus disease 2019 (COVID-19), caused by severe acute respiratory syndrome coronavirus 2 (SARS-CoV-2), is a major social and health concern worldwide. This pandemic has continued for over two years with the emergence of several variants of concern [1,2,3]. As of 13 May 2022, 6,259,945 cumulative deaths occurred globally [4], resulting in a significant impact on healthcare systems. South Korea reported 17,658,794 cumulative confirmed cases and 23,491 deaths in May of 2022 with variant outbreaks [5]. 

Certain evolving SARS-CoV2 variants have exhibited rapid emergence within populations. They show different transmissibility and clinical implications when compared with wild-type strains. The World Health Organization (WHO) assigned the Greek alphabet to these notable variants [6]. Omicron is a variant that was firstly reported in Botswana and shortly after in South Africa, in November 2021. The variant harbors over 30 mutations in the spike protein, which are related to enhanced transmissibility and reduced susceptibility to neutralizing antibodies [7]. Therefore, Omicron contributed to the majority of SARS-CoV-2 infections in many countries, posing a severe threat to global health care systems [8,9]. The Beta variant predominated in South Africa in late 2020 and is associated with immune evasion [10]. The Delta variant was firstly identified in India and was the most predominant variant globally until the emergence of Omicron. The Alpha variant was identified firstly in the United Kingdom in late 2020 and became dominant worldwide before the emergence of the Delta variant [11,12,13]. The pandemic caused by these variants persists until the time of writing. Therefore, the demand for rapid and accurate detection of SARS-CoV-2 remains high. 

Real-time reverse transcription-polymerase chain reaction (rRT-PCR) is widely used as the standard technique for detecting SARS-CoV-2 [14]. This method enables sensitive detection in specimens with low viral loads owing to the amplification of viral nucleic acids. However, this methodology requires competent laboratory personnel, facilities, and is a time-consuming process [15]. Therefore, rapid antigen tests (RATs) were developed and used as an alternative option for detecting SARS-CoV-2. RATs enable random access and shorter turnaround times without skilled workers or specialized infrastructure, allowing immediate isolation to prevent transmission [16]. In particular, instrument-free RATs have the highest usage in clinical settings because of their low cost and advantage of self-testing. However, reduced sensitivity compared with rRT-PCR is a limitation of RATs. Therefore, testing for suspected individuals with symptoms is recommended based on several guidelines [17,18]. 

RATs were widely utilized in the COVID-19 pandemic caused by diverse variants. Several reports for the diagnostic utility of RATs for variants showed discrepant results. Some studies showed acceptable performance of RATs for variants [19,20], whereas other studies demonstrated a lower performance of RATs for detecting variants of concern [21,22]. In addition, gene target failures, resulting in false negative results, were reported in commercially available rRT-PCR assays [9]. Information on the performance of these assays in the real world is urgently needed for decision makers. However, to the best of our knowledge, living systematic reviews and meta-analyses for RATs assayed for the detection of variants of SARS-CoV-2 are lacking. Therefore, this study aimed to evaluate and compare the available evidence on the real-world diagnostic utility of RATs for detecting variants at the point of care. In particular, the presence of symptoms or not are applied to the analysis. In addition, we focused on the instrument-free RATs most commonly used in actual clinical settings. 

## 2. Materials and Methods

### 2.1. Protocol and Registration

We developed, conducted, and reported this review following preferred reporting items for systematic review and meta-analysis of diagnostic test accuracy studies (PRISMA-DTA). The protocol was drafted and registered in the international prospective register for systematic reviews (PROSPERO), with the registration number: CRD42022329714 (Appendix A), before the systematic review and meta-analysis were conducted.

### 2.2. Eligibility Criteria

The eligibility criteria for studies were formulated based on the population, index test, reference test, and outcome for the review question. We included studies that evaluated the clinical performance of RATs for detecting SARS-CoV-2 variants in terms of sensitivity and specificity against rRT-PCR as the reference standard. The study population consisted of participants who had undergone testing for the detection of SARS-CoV-2. The description for the presence of symptoms or not was checked due to the importance of clinical presentation [16]. Original studies, including diagnostic cross-sectional and cohort studies, in English, published in the year 2020 and up until 10 April 2022, that described their methods and reported enough data for the construction of the standard two-by-two table were included.

### 2.3. Index Test

RATs are designed to directly detect SARS-CoV-2 viral proteins produced by the replicating virus. Among diverse methods, most self-testing RATs for COVID-19 utilize a sandwich immunological detection method, employing a simple-to-use lateral flow test format. In case of SARS-CoV-2, the N protein is frequently adopted as a target analyte owing to its expression levels in the early stage and because it displays the least amount of variation [23]. After collecting the respiratory specimen and applying it to the test strip, results are read by the operator within 10 to 30 min [24]. We included studies for instrument-free RATs, which have been most frequently used because of their cost effectiveness and accessibility, to reflect real-world usage as a point-of-care test. In addition, the specimens used in the included studies were nasal or nasopharyngeal samples, which are mostly recommended by the manufacturers of RATs [16].

### 2.4. Reference Standard

rRT-PCR is a molecular test that detects the genetic material (RNA) of SARS-CoV-2. It combines the laboratory techniques of reverse transcription and real-time polymerase chain reaction which amplifies specific complementary DNA (cDNA) targets. The number of amplification cycles required to reach the threshold level of detection is reported as the cycle threshold (Ct) value of rRT-PCR. Ct values are usually associated with viral loads. Among the currently available diagnostic tests for SARS-CoV-2, rRT-PCR is considered as the gold standard [16,24]. We have included studies presenting Ct values enabling the estimation of viral loads. The studies including less than 10% of specimens having Ct values more than 30 were excluded to prevent estimating over performance of RATs because of sample biases based on guidelines provided by the Korean Center for Disease Control and Prevention and the WHO [5,25]. 

### 2.5. Search Strategy

We systematically searched PubMED, Ovid-MEDLINE, Ovid-EMBASE, and CENTRAL, as well as the Korean databases (KMBASE), through to 29 June 2021. The search strategy included keywords such as “coronavirus infections”, “COVID-19”, “coronavirus disease 2019”, “COVID19” “SARS-CoV2”, “point-of-care testing”, and “rapid antigen test”. Ongoing trials and pre-published articles were excluded. A hand search through the reference lists of relevant primary and review articles was also performed for completeness. Since new evidence on the screening test is continuously produced, the search was updated on the 10th day of each month starting from September 2021 to 14 March 2022. We systematically searched Ovid-MEDLINE for the search updates. The electronic search strategy is presented in the Appendix A.

### 2.6. Study Selection and Data Extraction 

Studies searched by predefined formula were reviewed, independently, for eligibility by two reviewers (JK and SJ). After titles and abstracts were screened, full-text articles were assessed. JK and SJ rechecked the decisions and upon disagreements, all reviewers reviewed the inclusion and exclusion criteria and the decision for inclusion was made based on the majority decision. MC made the final decision after thorough reviewing when an agreement could not be reached.

All studies retrieved from our search strategies that matched our PICO questions and inclusion criteria were imported to Covidence (https://www.covidence.org/, accessed on 21 March 2022), a research tool to help living systematic review. The titles of the studies were firstly screened followed by the abstracts. Screened studies proceeded to the full-text review stage in the Covidence system based on the decisions of the reviewers. Full texts of all eligible studies were retrieved and assessed for eligibility. Final data were extracted from the included studies into an Excel spreadsheet. The following information was extracted (if available): demographic details of study participants, the total number of study participants, type of study, type of specimen used in RATs, and outcome measures, such as sensitivity and specificity. After extracting the data, JM and SR checked the extracted data. Upon disagreements, all reviewers reviewed the final extracted data, and disagreements were resolved through discussion via the majority decision.

### 2.7. Assessment of Risk of Bias and Applicability

We used the Quality Assessment of Diagnostic Accuracy Studies 2 (QUADAS-2) [26] tool to evaluate the methodological quality and applicability of the included studies at the study level. The quality assessment of each included study was performed by two reviewers (JM and SR). Any disagreements were resolved by MY. 

### 2.8. Statistical Analysis and Data Synthesis

We extracted data from the studies to construct standard two-by-two tables. The raw dataset for this systematic review (Appendix A) was deposited into the HARVARD Dataverse (https://doi.org/10.7910/DVN/5BMHBI, accessed on 26 May 2022). For each included study, diagnostic sensitivity and specificity with the 95% confidence interval (CI) were calculated based on the extracted two-by-two tables. Individual study participants were used as the unit of analysis throughout this work. If a study reported repeat testing of individuals, only the initial test was included in our analyses. Studies evaluating multiple types of RATs were considered as individual datasets. A random effects model with 95% CIs was used to calculate pooled sensitivity and specificity using Review Manager (RevMan, version 5.4, The Nordic Cochrane Centre, Copenhagen, Denmark). Higgins’ I^2^ was used to measure heterogeneity (I^2^ > 50% indicating substantial heterogeneity). For the hierarchical summary receiver operating characteristics (hsROC) curve, the parameters were obtained from the bivariate analysis using the “MetaDTA” site [27]. The parameter values were then added to the RevMan program. 

The studies were grouped based on (a) variants of concern; (b) clinical presentation at the time of testing such as symptomatic, asymptomatic, and mixed; (c) type of commercial RATs; and (d) the age of study population such as adults or children. Subgroup analyses were conducted in STATA 14.0 (StataCorp LP, College Station, TX, USA) using STATA “meta” according to the predefined subgroups. A descriptive analysis was performed and only accuracy ranges are reported when there were fewer than two studies. In addition, a sensitivity analysis for studies with different study populations was conducted to assess the potential bias. Furthermore, analyses including studies with sensitivity values only were also performed to overcome the deviation of estimated results stemming from the lack of included studies.

## 3. Results

### 3.1. Identified Studies

We found 2537 records after performing a systematic search for articles, resulting in 1796 records after excluding duplicates. After seven update searches, the titles and abstracts of 2398 records were reviewed and screened. A total of 1813 studies were excluded because they did not meet the selection criteria. Full-text screening of the remaining 585 studies yielded six studies that met the eligibility criteria. The final identified studies were included in this systematic review. The preferred reporting items for the systematic review and meta-analysis (PRISMA) flow diagram for the selection of studies is depicted in Figure 1.

### 3.2. Study Characteristics

The six included studies included 7562 study participants and were published in 2022, except for one study published in 2021 for the Beta variant. Alpha, Beta, Delta, and Omicron variants were covered by these studies. Most of the studies (five out of six) dealt with the combination of symptomatic and asymptomatic participants. Symptomatic patients suspected of having COVID-19 were the study population in four studies, while another three studies included asymptomatic individuals as the study population. Nasopharyngeal swabs rather than nasal swabs were commonly used for RATs. The applied RATs and rRT-PCR assays were totally different among the included studies. Commercially available RATs, such as BioSpeedia COVID19 Nasal Antigen Test (BioSpeedia, Gutenberg, France), PanBio SARS-CoV-2 RTD (Abbott, Chicago, IL, USA), STANDARD Q COVID-19 Ag Test (SD Biosensor, Suwon, Korea), QuickNavi-Flu+COVID19 Ag (Denka Co., Tokyo, Japan), and BinaxNOW COVID-19 Antigen Self-Test (Abbott, Chicago, IL, USA) were used. The characteristics of the included studies are presented in Table 1.

### 3.3. Methodological Quality of Studies 

Most studies had an unclear risk of patient selection bias. All studies avoided case-control design; however, there were insufficient descriptions about the inappropriate exclusion of patients and non-random sampling of patients. The two studies [19,28] utilizing prospectively collected or routine samples were classified as having a low risk of bias. Regarding applicability, low risks were determined for the included studies because the study population of all studies could be the predefined population of the review question. With regard to the index test, the RATs in all the studies were conducted immediately after sample collection and interpreted without knowledge of the reference standard, indicating a low risk of bias. Five studies had a low risk for applicability concerns of the index test, while one study [29] was designated as having an unclear risk; the study did not provide detailed information about the manufacturer. 

Regarding the reference standard risk of bias, two studies [29,30] had a high risk of bias because the results of the RATs were told to the participants before the rRT-PCR was performed. Unclear risks of applicability concerns were assigned to two studies [20,31] adopting in-house rRT-PCR assays. Under the flow and timing domain, sampling and processing for the index test and reference standard were performed at almost the same time in all the included studies. In addition, all patients received the same reference standard and were included in the analysis, leading to a low risk of bias with regard to patient flow. The methodological quality of the included studies is illustrated in Figure 2 and Figure 3.

### 3.4. Meta-Analysis of RATs for Detecting Variants 

A total of 7562 samples assayed using RATS for variants and confirmed with rRT-PCR as a reference standard were included in this meta-analysis. The pooled sensitivity and specificity of RATs for variants of concern were 69.7% (95% CI = 62.5% to 76.1%) and 100.0% (95% CI = 98.8% to 100.0%), respectively (Figure 4). The heterogeneity of the included studies was observed based on the hsROC curve. As a result of examining the diagnostic accuracy of RATs according to the variants (Table 2), the sensitivity of Alpha ranged from 69.6% to 76.4% and that of Delta was from 57.3% to 80.9%. The studies on Delta and Omicron showed 61.5% sensitivity. 

### 3.5. Analysis Stratified by the Presence of Symptoms

Subgroups defined by the clinical presentation at the time of testing showed pooled sensitivity ranging from 79.2% (95% CI = 69.7% to 86.3%) (for tests performed on symptomatic patients) to 58.0% (95% CI = 40.3% to 73.9%) (for tests performed on only asymptomatic patients) (Figure 5, Figure 6 and Figure 7). For symptomatic patients, the sensitivities of RATs were 82.9% for Alpha, 69.2% for Beta, 73.7% for a nasal swab for Delta, and 88.3% for a nasopharyngeal swab for Delta (Table 2). In terms of asymptomatic participants, the sensitivity of Alpha was 27.3%. Those for Delta were lower (60.0% for nasal swab and 69.4% for nasopharyngeal swab) when compared with reported sensitivities in symptomatic patients. 

### 3.6. Sensitivity Analysis

A study evaluating RATs for the Delta or Omicron variants in fully vaccinated participants showed the lowest sensitivity (57.3%) [28]. When the data in this study were excluded, the estimated sensitivity (71.7%) marginally increased without a statistical difference. Regarding the study population, a study investigating the performance of RATs in children aged 0 to 15 years exhibited a sensitivity of 69.6%. The pooled sensitivity was not significantly different (69.6%) from that of the overall analysis (69.7%) when the dataset of this study was subtracted.

### 3.7. Analysis of RATs in Studies with Sensitivity Only

The seven studies reporting sensitivity without specificity were analyzed additionally because of the small number of included studies of RATs for variants (Figure 8). The pooled sensitivity (50.0%, 95% CI = 44.0% to 55.0%) was lower than that of the overall analysis of the formally selected studies. When the values of the additional and formally identified studies were combined, the summary sensitivity was 53.0% (95% CI = 47.0% to 59.0%) (Figure 9).

## 4. Discussion

In this study based on a living systematic review and meta-analysis, the evidence of the diagnostic utility of RATs without instruments for SARS-CoV-2 variants of concern were combined and compared according to the presence of symptoms. The pooled sensitivity and specificity of RATs for variants from 6904 participants in six studies [19,20,28,29,30,31] were 69.7% and 100.0%, respectively. The reported sensitivities of symptomatic patients were higher than those of asymptomatic individuals. 

There have been several systematic reviews for the diagnostic accuracy of RATs for detecting wild type SARS-CoV-2. The pooled sensitivities of these meta-analyses ranged from 56.2% (95% CI = 29.5% to 79.8%) to 82% (95% CI = 71% to 89%) [16,24,32,33,34,35,36,37]. The advancement of developed RATs, the target study population, types of RATs, and search timing could influence the pooled diagnostic sensitivity values. Our pooled sensitivity for variants exhibited a relatively low value (69.7%) when compared with those of previous articles for wild type SARS-CoV-2. However, it was within the 95% CI of pooled sensitivities in most of the previously published meta-analyses [16,24,32,34,35,36,37], evidencing the continuous use of RATs without extra interventions in the COVID-19 pandemic caused by SARS-CoV-2 variants. In terms of specificity, the pooled specificities were consistently high (98% to 100%), which was similar to those in our study (100.0%).

Based on the guidelines recommended by WHO, the use of RATs should be prioritized for symptomatic patients, as well as asymptomatic individuals at a high risk of COVID-19, especially in cases with a limited capacity for rRT-PCR [16]. The presence of symptoms is so important that it is also used in the criteria for discharging patients from isolation [38]. Therefore, we analyzed the sensitivity and specificity of RATs considering the presence of symptoms. RATs exhibited consistently high specificity, regardless of symptoms. However, the sensitivities of RATs for variants in symptomatic patients were higher than those in asymptomatic individuals. The substantially higher sensitivities in symptomatic patients than those in asymptomatic participants were consistently reported according to the previous systematic reviews without separation of variants [16,35]. A systematic review with electronic searches on 30 September 2020 [35] showed a pooled sensitivity of 72.0% in symptomatic patients and 58.1% in asymptomatic individuals without any overlapping of the 95% CI. Another review on 26 August 2021 [16] also demonstrated a significant difference in the pooled sensitivities between symptomatic (82%) versus asymptomatic (68%) participants. The increased chance of higher viral loads in symptomatic patients rather than asymptomatic individuals could be the cause of these findings [24].

The WHO and U.S. Food and Drug Administration (FDA) recommend a minimum sensitivity of more than 80% [39,40]. Based on these criteria, the overall use of RATs is not recommended based on our pooled sensitivity of RATs for variants. Regarding each assay, the BinaxNOW COVID-19 Ag Self-Test (Abbott, Jena, Germany) exhibited a pooled sensitivity of 79% according to a recent meta-analysis [16]. Meanwhile, the BinaxNOW COVID-19 Ag Self-Test showed a 61.5% sensitivity for the detection of Delta and Omicron variants of SARS-CoV-2 [31]. Omicron (B.1.1.529 lineage) is predominantly attributed to the COVID-19 pandemic [8]. The clinical impact of this variant stems from a greater replication rate than the Delta variant [41] and an immune escape from humoral immunity. The ratio of reinfection to primary infection was higher during the Omicron surge when compared with Beta and Delta variants [42]. The Delta (B.1.617.2 lineage) variant, the most commonly detected variant before the emergence of Omicron, exhibited more transmissibility [43,44] and was related to a greater severity of COVID-19 symptoms and hospitalization [44,45,46,47] when compared with the Alpha variant (B.1.1.7 lineage). The STANDARD Q COVID-19 Ag Home Test (SD Biosensor, Seoul, South Korea) showed a low sensitivity of 57.3% in a report studied during the Delta wave in fully vaccinated participants [28]. According to a systematic review, the STANDARD Q COVID-19 Ag Home Test was frequently used for several studies and its pooled sensitivity was 74.9% (95% CI = 69.3% to 79.7%) [24]. The performance of the Panbio COVID-19 Ag RDT (Abbott, Jena, Germany) was also commonly reported and its pooled sensitivity was 71.8% (95% CI = 65.4% to 77.5%) based on a systematic review published in 2021 [24], and 75.9% according to a meta-analysis published in 2022 [16]. Among the included studies in our meta-analysis, a study investigating the performance of Panbio COVID-19 Ag RDT for the detection of the Beta variant showed a relatively low sensitivity (69.2%) [30]. Based on these findings, analyses stratified by applied assays are necessary to assess the performance of RATs for variants.

We have included studies providing values for sensitivity and specificity concurrently because separate pooling is not recommended for a systematic review and meta-analysis [48]. However, there were several studies evaluating the sensitivity of RATs for variants without specificity because the sensitivity is an important variable considering RATs as screening tests rather than confirming assays. Therefore, we searched manually and reviewed an additional seven studies with sensitivities for variants [21,22,49,50,51,52,53], which were not included in the formal meta-analysis. Five out of seven studies dealt with the Delta and Omicron variants. When the pooled sensitivity was calculated separately, it was 50.0% (53.0% for combined 13 studies), indicating a reduced diagnostic utility in RATs.

Among the included studies, a study investigating the reliability of a RAT for Delta or Omicron variants in fully vaccinated participants showed the lowest sensitivity of 57.3% [28]. Relatively low viral loads in vaccinated individuals when compared with those in unvaccinated participants [54,55] could be the cause of these findings. When the presence of symptoms was considered, a study evaluating the RAT in a pediatric emergency department [19] exhibited the lowest sensitivity (27.3%) for asymptomatic children. According to a recent meta-analysis for children [56], the broad implementation of RATs was not recommended because of low diagnostic sensitivity.

Regarding quality assessment, most studies showed a low risk of bias and applicability concerns based on the QUADAS-2 assessment despite slight heterogeneity. Meanwhile, unclear patient selection was derived from insufficient details about the selection criteria and non-random sampling. Due to the high risk of the reference standard, the results of the reference standard were interpreted with knowledge of the index tests in some studies.

Several limitations should be considered in this study. First, heterogeneity of included studies stemming from study populations, the timing of sampling collection, and applied RATs and rRT-PCR assays might affect the results of this meta-analysis. Subgroup analyses and a random effect model were conducted to overcome this limitation [57,58]. Second, publication bias could not be evaluated well because of the lack of included studies and techniques for the detection of the accuracy of diagnostic tests. Further qualified studies conducted by diverse and independent research teams will reduce this risk. Third, the detailed subgroup analyses stratified by types of variants and applied RATs were not applicable due to the limited number of studies. Additional updates on the strength of living guidelines [59] could overcome these limits. Further studies performing the evaluation of the diagnostic utility of RATs for variants of concern are necessary to provide detailed information about RATs.

## 5. Conclusions

In summary, our living systematic meta-analysis demonstrated that the diagnostic utility of the instrument-free RATs for variants including Alpha, Beta, Delta, and Omicron was insufficient according to the criteria recommended by WHO and FDA, especially in the sensitivity aspect. The sensitivities of symptomatic patients were greater than those of asymptomatic participants. To the best of our knowledge, this is the first study reporting the specific pooled values for the performance of RATs for SARS-CoV-2 variants based on a living systematic review and meta-analysis to facilitate appropriate diagnosis and management of COVID-19 patients. Based on our findings, reassessment and monitoring of the diagnostic utility of RATs for variants are required. In addition, diagnostic screening nucleic acid amplification tests could be an alternative method for detecting variants. Subsequent whole genome sequencing or at least complete or partial S-gene sequencing are recommended to confirm the identification of variants of concern. Laboratories should remain vigilant to the performances of adopted assays during the COVID-19 pandemic caused by variants.

## Figures and Tables

**Figure 1 viruses-14-01479-f001:**
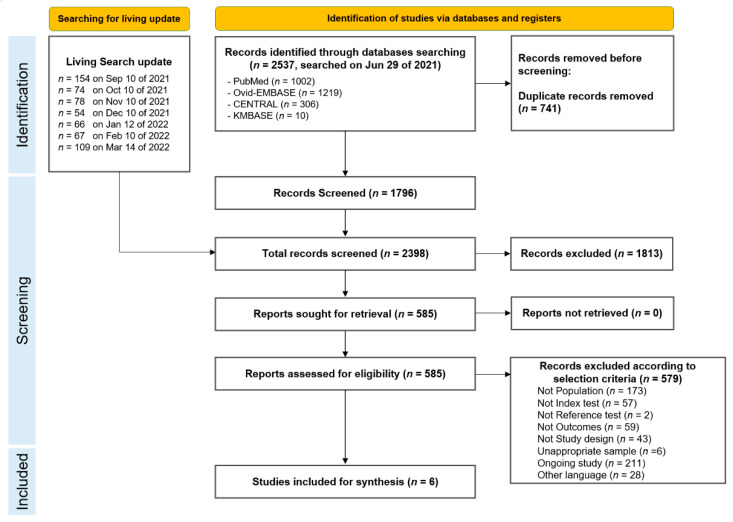
The preferred reporting items for systematic review and meta-analysis (PRISMA) flow diagram.

**Figure 2 viruses-14-01479-f002:**
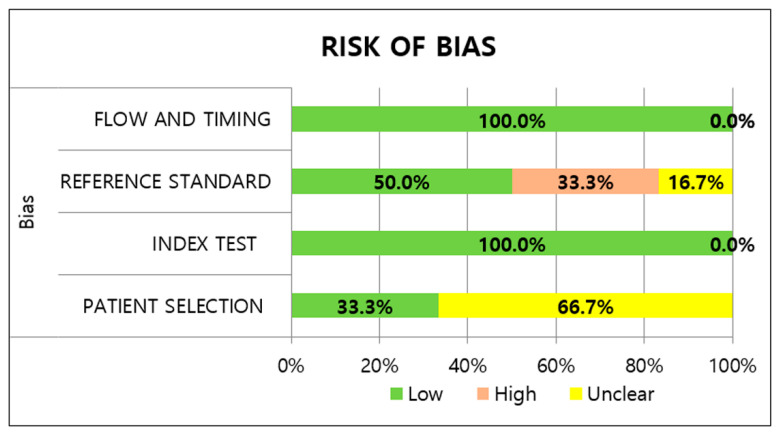
Methodological quality of included studies for risk of bias.

**Figure 3 viruses-14-01479-f003:**
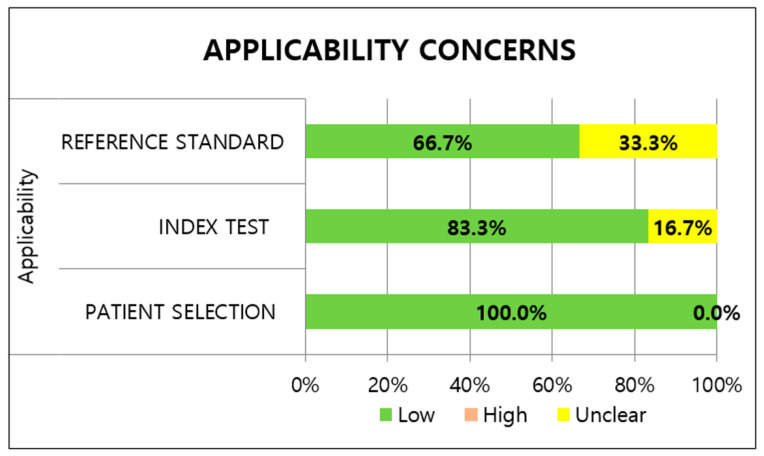
Methodological quality of included studies for applicability.

**Figure 4 viruses-14-01479-f004:**
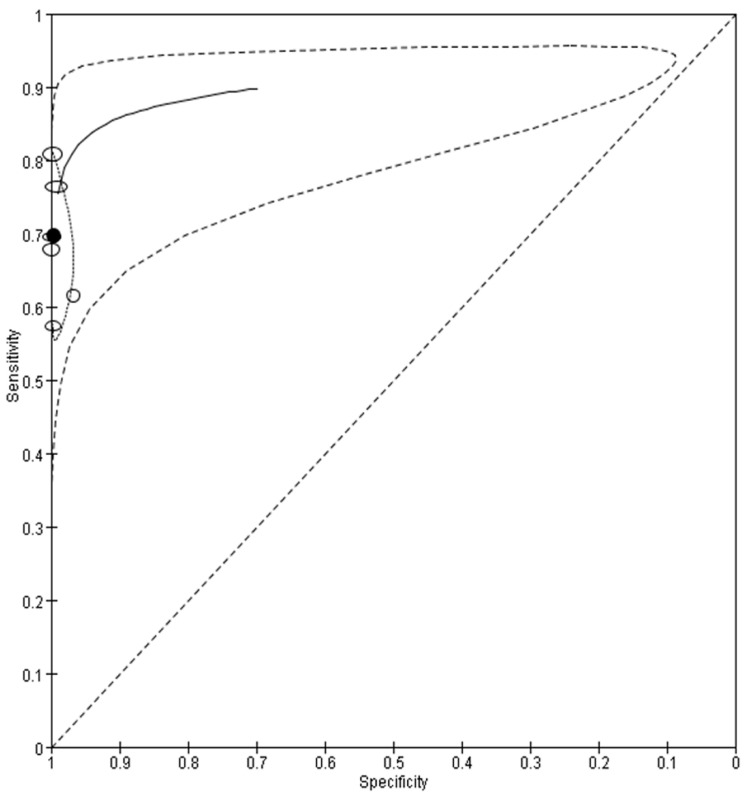
Hierarchical summary receiver operating characteristics curve of rapid antigen tests for variants. Summary point indicates the pooled values for sensitivity and specificity. Black line is the summarized receiver operating characteristics curve. Black dot indicates the pooled sensitivity and specificity of included studies. The size of the circle is proportional to the size of the study.

**Figure 5 viruses-14-01479-f005:**
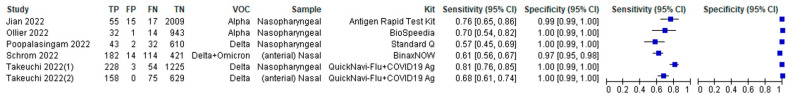
Coupled forest plot of sensitivity and specificity: mixed participants. TP, true positive; FP, false positive; FN, false, negative; TN, true negative, VOC, variants of concern.

**Figure 6 viruses-14-01479-f006:**
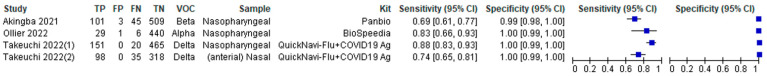
Coupled forest plot of sensitivity and specificity: symptomatic patients. TP, true positive; FP, false positive; FN, false, negative; TN, true negative, VOC, variants of concern.

**Figure 7 viruses-14-01479-f007:**
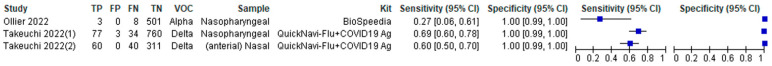
Coupled forest plot of sensitivity and specificity: asymptomatic individuals. TP, true positive; FP, false positive; FN, false, negative; TN, true negative, VOC, variants of concern.

**Figure 8 viruses-14-01479-f008:**
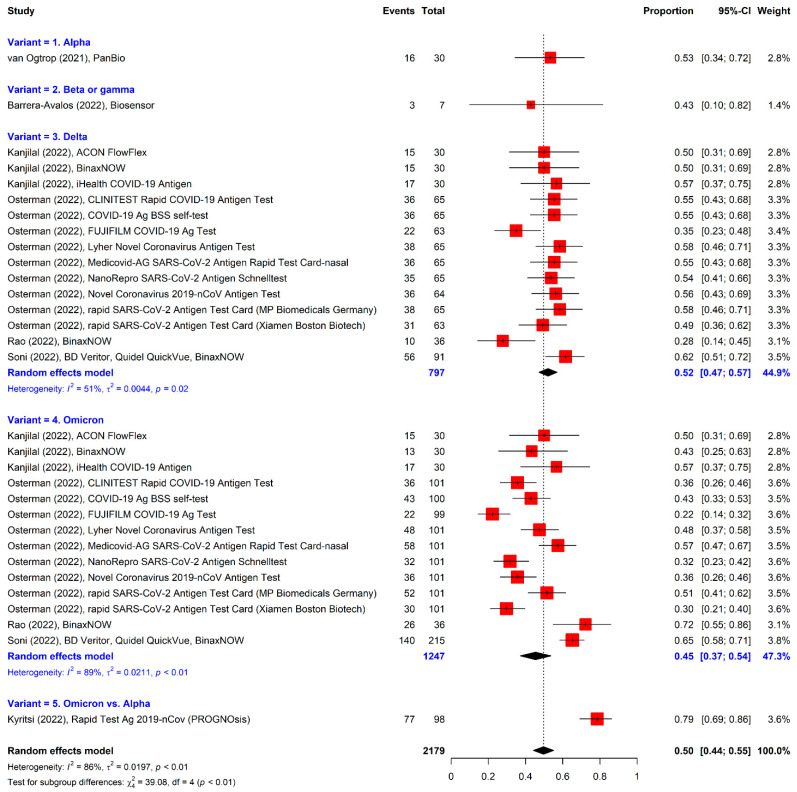
Forest plot using univariate analysis only for the sensitivity of RATs.

**Figure 9 viruses-14-01479-f009:**
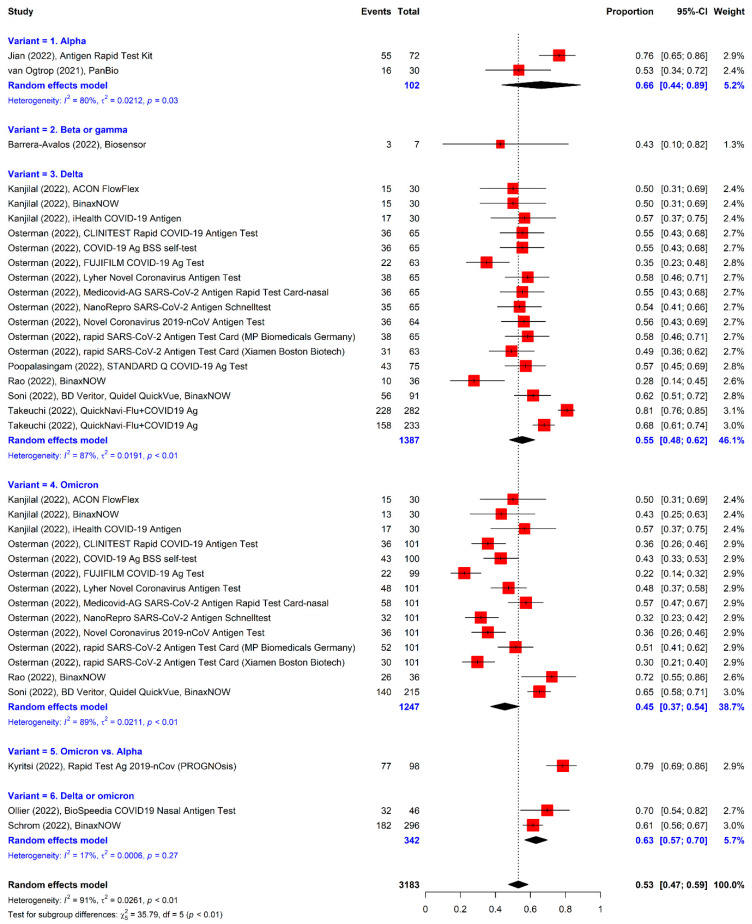
Forest plot for combining the formal and additional studies of RATs. The estimated pooled values of subgroups are presented in blue color.

**Table 1 viruses-14-01479-t001:** Characteristics of included studies stratified by variants.

Study	Study Population (*n*)	Enrollment Periods	Sample Type	RATs	rRT-PCR	Study Location
Alpha						
Jian et al. [2022]	Mixed (2096)	2021.5.17~2021.5.22	NP swab	Antigen Rapid Test Kit	LabTurbo AIO 48 system (LabTurbo Biotech Corporation, Princeton, NJ, USA)	Taiwan
Ollier et al. [2022]	Mixed (990)	2021.1.15~2021.5.28	NP swab	BioSpeedia COVID19 Nasal Antigen Test (BioSpeedia, Gutenberg, France)	SARS-CoV-2 R-GENE (bioMérieux, Marcy I’Etoile, France)	France
Beta						
Akingba et al. [2021]	Symptomatic (677)	2020.11.17~2020.11.20	NP swab	PanBio SARS-CoV-2 RTD (Abbott, Chicago, IL, USA)	Allplex 2019-nCoV (Seegene, Seoul, South Korea)	South Africa
Delta						
Poopalasingam et al. [2022]	Mixed (696)	2021.12.6~2021.12.31	NP swab	STANDARD Q COVID-19 Ag Test (SD Biosensor, Suwon, Korea)	cobas SARS-CoV-2 (Roche Diagnostics, Mannheim, Germany)	Germany
Takeuchi et al. [2022]	Mixed (2372)	2021.8.2~2021.9.13	NP and Nasal swabs	QuickNavi-Flu+COVID19 Ag (Denka Co., Tokyo, Japan)	In-house	Japan
Delta and Omicron						
Schrom et al. [2022]	Mixed (731)	2022.1	Nasal swab	BinaxNOW COVID-19 Antigen Self Test (Abbott, Chicago, IL, USA)	In-house	USA

RATs, rapid antigen tests; rRT-PCR, real-time reverse transcription-polymerase chain reaction; Mixed, symptomatic and asymptomatic; NP, nasopharyngeal.

**Table 2 viruses-14-01479-t002:** Summary diagnostic accuracy data for RATs according to the clinical presentation.

Clinical Presentation	Study	Sample Type	RATs	Sensitivity (%) *	Specificity (%) *	Study Location
Mixed	Alpha					
	Jian et al. [2022]	NP swab	Antigen Rapid Test Kit	76.4 (64.9–85.6)	99.3 (98.8–99.6)	Taiwan
	Ollier et al. [2022]	NP swab	BioSpeedia COVID19 Nasal Antigen Test (BioSpeedia, Gutenberg, France)	69.6 (54.3–82.3)	99.9 (99.4–100.0)	France
	Delta					
	Poopalasingam et al. [2022]	NP swab	STANDARD Q COVID-19 Ag Test (SD Biosensor, Suwon, Korea)	57.3 (46.1–67.9)	99.9 (99.6–100.0)	German
	Takeuchi et al. [2022]	NP swab	QuickNavi-Flu+COVID19 Ag (Denka Co., Tokyo, Japan)	80.9 (75.8–85.3)	99.8 (99.3–99.9)	Japan
	Takeuchi et al. [2022]	Nasal swab	QuickNavi-Flu+COVID19 Ag (Denka Co., Tokyo, Japan)	67.8 (61.4–73.8)	100 (99.1–100.0)	Japan
	Delta and Omicron					
	Schrom et al. [2022]	Nasal swab	BinaxNOW COVID-19 Antigen Self Test (Abbott, Chicago, IL, USA)	61.5 (55.7–67.1)	96.8 (94.7–98.2)	USA
Symptomatic	Alpha					
	Ollier et al. [2022]	NP swab	BioSpeedia COVID19 Nasal Antigen Test (BioSpeedia, Gutenberg, France)	82.9 (66.4–93.4)	99.8 (98.7–100.0)	France
	Beta					
	Akingba et al. [2021]	NP swab	PanBio SARS-CoV-2 RTD (Abbott, Chicago, IL, USA)	69.2 (61.4–75.8)	99.0 (98.8–99.3)	South Africa
	Delta					
	Takeuchi et al. [2022]	NP swab	QuickNavi-Flu+COVID19 Ag (Denka Co., Tokyo, Japan)	88.3 (82.5–92.7)	100.0 (98.8–100.0)	Japan
	Takeuchi et al. [2022]	Nasal swab	QuickNavi-Flu+COVID19 Ag (Denka Co., Tokyo, Japan)	73.7 (65.3–80.9)	100.0 (98.3–100.0)	Japan
Asymptomatic	Alpha					
	Ollier et al. [2022]	NP swab	BioSpeedia COVID19 Nasal Antigen Test (BioSpeedia, Gutenberg, France)	27.3 (6.1–61.0)	100.0 (99.3–100.0)	France
	Delta					
	Takeuchi et al. [2022]	NP swab	QuickNavi-Flu+COVID19 Ag (Denka Co., Tokyo, Japan)	69.4 (59.9–77.8)	99.6 (98.9–99.9)	Japan
	Takeuchi et al. [2022]	Nasal swab	QuickNavi-Flu+COVID19 Ag (Denka Co., Tokyo, Japan)	60.0 (49.7–69.7)	100.0 (98.2–100.0)	Japan

* Sensitivity and specificity are expressed as values (95% confidence interval). RATs, rapid antigen tests; Mixed, symptomatic and asymptomatic; NP, nasopharyngeal.

## Data Availability

All data used and presented in this study are deposited in the HARVARD Dataverse (https://doi.org/10.7910/DVN/5BMHBI).

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
