# Peer review of "Clinical Performance of Rapid and Point-of-Care Antigen Tests for SARS-CoV-2 Variants of Concern: A Living Systematic Review and Meta-Analysis"

_viruses, 2022, doi:10.3390/v14071479_

Round 1

Reviewer 1 Report

Overall a well written review on Clinical Performance of Rapid and Point-of-Care Antigen.

The manuscript has been very well written.

RATs are of importance for the rapid detection of severe acute respiratory syndrome coronavirus 2 (SARS-CoV-2)

Good amount of the samples are presented in the manuscript .Testing  of the diagnostic utility of RATs for variants of concerns will be needed for additional improvement.

Author Response

We are very appreciated with your comments for this manuscript. As suggested by the reviewer, direct evaluation of rapid antigen tests (RATs) for variants of concerns is necessary. There have been several studies for the performance of RATs detecting variants of concerns and further large studies evaluating RATs for variants of concerns are planned by the members of Korean Society for Laboratory Medicine and Korean Ministry of Food and Drug Safety, as far as I know. Therefore, we have added the necessity of studies for the evaluation of RATs for variants of concerns to the revised Discussion section (Page 17, lines 130 to 132) as follows.

Further studies performing the evaluation of the diagnostic utility of RATs for variants of concerns are necessary to provide detailed information about RATs.”

Reviewer 2 Report

The present study is well represented and written. However,  please consider the following changes before the publication 

In the introduction,  to explain the Omicron variant and other variants significance,  cite authentic recent publications after the 6 and 7 references such as :

10.1016/j.amsu.2022.103737

10.1016/j.envres.2022.112816

2. The conclusion must be improved  while providing the reliable alternatives methods to detect the infection caused by variants. 

3. Check the whole manuscript for the abbreviations , once it's provided then use the abbreviations. 

4. In the introduction the recent information about the effects of variants on the diagnostic methods is unsufficient and must be improved.

Author Response

  1. In the introduction, to explain the Omicron variant and other variants significance, cite authentic recent publications after the 6 and 7 references such as:

10.1016/j.amsu.2022.103737

10.1016/j.envres.2022.112816

à As suggested by the reviewer, we have revised and cited the references in the revised introduction section (Page 3, lines 1 to 7) as follows.

Omicron is a variant that was firstly reported in Botswana and then South Africa shortly after in November 2021. The variant harbors over 30 mutations in the spike protein, which are related to the enhanced transmissibility and reduced susceptibility to neutralizing antibodies [7]. Therefore, Omicron has contributed to the majority of SARS-CoV-2 infections in many countries posing a severe threat to global health care systems [8,9].”

  1. Islam, F.; Dhawan, M.; Nafady, M.H.; Emran, T.B.; Mitra, S.; Choudhary, O.P.; Akter, A. Understanding the omicron variant (B.1.1.529) of SARS-CoV-2: Mutational impacts, concerns, and the possible solutions. Ann Med Surg (Lond) 2022, 78, 103737, doi:10.1016/j.amsu.2022.103737.
  2. Khandia, R.; Singhal, S.; Alqahtani, T.; Kamal, M.A.; El-Shall, N.A.; Nainu, F.; Desingu, P.A.; Dhama, K. Emergence of SARS-CoV-2 Omicron (B.1.1.529) variant, salient features, high global health concerns and strategies to counter it amid ongoing COVID-19 pandemic. Environ Res 2022, 209, 112816, doi:10.1016/j.envres.2022.112816.

  1. The conclusion must be improved while providing the reliable alternatives methods to detect the infection caused by variants.

à We have inserted detailed recommendations and alternative methods for variants into the revised Conclusions section (Page 17, lines 141 to page18, lines 147) as follows.

“The reassessment and monitoring of the diagnostic utility of RATs for variants are required based on our findings. In addition, diagnostic screening nucleic acid amplification tests could be alternative methods for detecting variants. Subsequent whole genome sequencing, or at least complete or partial S-gene sequencing are recommended to confirm the identification of variants of concerns. Laboratories should be remained vigilant to the performances of adopted assays during COVID-19 pandemic caused by variants.”

  1. Check the whole manuscript for the abbreviations, once it's provided then use the abbreviations. 

 à We have checked the abbreviations and corrected the following words throughout the revised manuscript.

“RT-qPCR to rRT-PCR” and “World health Organization to WHO”

  1. In the introduction the recent information about the effects of variants on the diagnostic methods is unsufficient and must be improved

 à Based on the recommended references and our data, we have added the recent information about the effects of variants on RATs and rRT-PCR assays to the revised Introduction section (Page 3, lines 32 to 38) as follows.

 “Several reports for the diagnostic utility of RATs for variants showed discrepant results. Some studies showed acceptable performance of RATs for variants [19,20] whereas other studies demonstrated the lowered performance of RATs for detecting variants of concerns [21,22]. In addition, gene target failures, resulting in false negative results, reported in commercially available rRT-PCR assays [9].”
